# The Latest Approach of Immunotherapy with Endosomal TLR Agonists Improving NK Cell Function: An Overview

**DOI:** 10.3390/biomedicines11010064

**Published:** 2022-12-27

**Authors:** Irene Veneziani, Claudia Alicata, Lorenzo Moretta, Enrico Maggi

**Affiliations:** 1Translational Immunology Unit, Bambino Gesù Children’s Hospital, IRCCS, 00146 Rome, Italy; 2Tumor Immunology Unit, Bambino Gesù Children’s Hospital, IRCCS, 00146 Rome, Italy

**Keywords:** cancer immunotherapy, endosomal Toll-like receptors, natural killer cells, Toll-like receptor agonists

## Abstract

Toll-like receptors (TLRs) are the most well-defined pattern recognition receptors (PRR) of several cell types recognizing pathogens and triggering innate immunity. TLRs are also expressed on tumor cells and tumor microenvironment (TME) cells, including natural killer (NK) cells. Cell surface TLRs primarily recognize extracellular ligands from bacteria and fungi, while endosomal TLRs recognize microbial DNA or RNA. TLR engagement activates intracellular pathways leading to the activation of transcription factors regulating gene expression of several inflammatory molecules. Endosomal TLR agonists may be considered as new immunotherapeutic adjuvants for dendritic cell (DC) vaccines able to improve anti-tumor immunity and cancer patient outcomes. The literature suggests that endosomal TLR agonists modify TME on murine models and human cancer (clinical trials), providing evidence that locally infused endosomal TLR agonists may delay tumor growth and induce tumor regression. Recently, our group demonstrated that CD56^bright^ NK cell subset is selectively responsive to TLR8 engagement. Thus, TLR8 agonists (loaded or not to nanoparticles or other carriers) can be considered a novel strategy able to promote anti-tumor immunity. TLR8 agonists can be used to activate and expand in vitro circulating or intra-tumoral NK cells to be adoptively transferred into patients.

## 1. Introduction

Natural killer (NK) cells are key effectors of the innate immunity, which cooperate with adaptive immunity in the protection from microbial infections, viruses, fungi, and cancer. Innate immune cells have been initially considered unable to identify and eliminate microbes without pre-sensitization; however, several studies clearly demonstrated that innate immunity recognizes microbial-associated or pathogen-associated molecular patterns (PAMPs) through their pattern recognition receptors (PRRs), including Toll-like receptors (TLRs), NOD-like receptors (NLRs), C-type lectin receptors (CLRs), and RIG-I-like receptors (RLRs) [1,2]. Toll receptors were first described in the mid-1990s as essential molecules for embryonic patterning in drosophila that also play a role in antifungal immunity [3]. Subsequently a homologous family of Toll receptors, the so-called TLRs, were found in mammals [4]. TLRs are currently the most well-defined PRRs with respect to PAMP recognition and induction of innate immune response [5].

TLRs are type I transmembrane proteins with a N-terminal leucine-rich repeat (LRRs) ectodomains that mediate PAMP recognition, a transmembrane domain, and a C-terminal Toll-interleukin 1 (IL-1) receptor (TIR) domain necessary for signal transduction [6]. TIR domains are characteristic of many adaptor proteins that interact homo-typically with the TIR domains of TLRs and IL-1 receptors as the first step in the signaling cascade. Interestingly, homologs of TIR domains are also present in some plant proteins where they confer resistance to pathogens [7]; this suggests that the TIR domain represents a very ancient motif that served an immune function before the divergence of plants and animals. A typical stretch of around 20 hydrophobic residues composes the TLR transmembrane domain. In particular, endosomal TLRs transmembrane domains include the UNC93B protein that recognizes nucleic acid PAMPs and directs these TLRs to endocytic compartments [8]. The N-terminal ectodomains (ECDs) of TLRs are glycoproteins with 550–800 amino acid residues [9].

The TLR family comprises 10 members (TLR1/TLR10) in humans and 12 members in mice (TLR1-TLR9 and TLR11-TLR13), TLR1–TLR9 being conserved in both species [10]. The TLR genes are dispersed throughout the human genome: those encoding TLR1 and TLR6 map to human chromosome 4p14, TLR2 and TLR3 to 4q31.3–q35, TLR4 to 9q32–q33, TLR5 to 1q33.3–q42, TLR7 and TLR8 to Xp22, and TLR9 to 3p21.3.

Based on their cellular localizations, human TLRs can be divided in cell surface receptors (TLR1, 2, 4, 5, 6 and 10), which are primarily designated to recognize extracellular macromolecular ligands from bacteria and fungi, and endosomal (TLR3, 7, 8 and 9), recognizing cell ligands that require internalization to generate a signal as microbial DNA or RNA [11]. In particular, the TLR2-TLR1 heterodimer recognizes triacyl lipopeptides from Gram-negative bacteria and mycoplasma, whereas the TLR2-TLR6 heterodimer recognizes diacyl lipopeptides from Gram-positive bacteria and mycoplasma [12,13]. TLR4 responds to lipopolysaccharide, a surface structure of Gram-negative bacteria that can cause septic shock, whereas TLR5 binds the flagellin in bacterial flagella [14,15]. Among endosomal TLRs, TLR3 recognizes double-stranded RNA (dsRNA) produced by a number of replicating viruses [16,17,18]. By contrast, TLR7 and TLR8 recognize single-stranded RNA (ssRNA) derived from RNA of viruses, such as vesicular stomatitis virus, HIV, influenza A and some silencing RNAs [19,20,21]. Lastly, TLR9 recognizes unmethylated 2′-deoxyribo CpG DNA motifs in bacteria and viruses [5].

In addition to responding to PAMPs, TLRs respond to danger-associated molecular patterns (DAMPs), also called alarmins, and trigger inflammatory responses. Alarmins are produced as a result of cell death and injury or by tumor cells [22,23]. Thus, TLRs are critical sensors for immunosurveillance against tumors. The tumor microenvironment (TME) is rich in molecules potentially able to activate TLR signaling in local antigen presenting cells (APCs) to improve anti-tumor T cell responses, such as heat shock proteins, high mobility group proteins, DNA from necrotic cells, and hyaluronic acid [24,25]. However, besides their role in inducing anti-tumor response, tumor cells may activate negative regulatory circuits critical for normal homeostasis of the immune system through TLRs by inducing and maintaining immune tolerance to cancer. The most active negative regulators include extracellular decoy receptors (soluble TLRs), transmembrane suppressive receptors, several miRNAs, and intracellular inhibitors [26]. After ligand engagement, TLRs form homodimers or heterodimers and undergo conformational changes to recruit downstream adaptor proteins. Examples are the TLR3 homodimer that recruits the TIR domain-containing adaptor inducing IFN-β (TRIF), the homodimer TLR9 and the heterodimer TLR7/8, both recruiting myeloid differentiation primary response gene 88 (MyD88). Both pathways lead to activation of transcription factors, such as NF-kB, AP1, CREB, IRF3/7, which regulate gene expression of type I and III IFN, inflammatory cytokines/chemokines, costimulatory and adhesion molecules, and antimicrobial mediators [27]. TLR expression and function have been widely studied in APCs, but some reports provided clear evidence that they may be the first-line defense also on NK cells against bacterial, viral, and fungal pathogens. In addition, TLR ligands can activate NK cells directly or indirectly with accessory cells through cytokines or cell-to-cell contact [27,28,29], therefore, they have the potential to stimulate immunological effector function of NK cells for cancer immunotherapy and infectious diseases. In this review, we will discuss the features of endosomal TLRs in human NK cells and the role of their agonists in immunotherapy.

## 2. TLR Expression in NK Cells

Controversial observations have been reported on the expression of TLRs (especially of endosomal TLRs) on human NK cells, probably due to no specific detecting antibodies. However, the use of quantitative PCR primers specifically discriminating among TLR family members allowed to detect mRNA generating reliable data about the presence of each TLR in different cell types [11,27,30,31].

Data report that TLR family members are expressed in immune cells, but also in a variety of other cells, including vascular endothelial cells, adipocytes, cardiac myocytes, and intestinal epithelial cells. The expression level of TLRs among immune cells is variable [32]. During the last two decades, the group of Alessandro and Lorenzo Moretta highlighted and repeatedly confirmed the presence of all endosomal TLRs on NK cells by different methods (mRNA expression, Western Blot analysis, and confocal microscopy) performed on purified NK cells, NK92 cell lines, or NK cell clones [33,34,35,36,37]. Recently, the presence of endosomal TLRs has been directly demonstrated at the protein level and functionally through the activation of purified NK cells by specific ligands/agonists, particularly that of TLR8 [38]. NK cells express all endosomal TLRs independently of their state of activation at different levels. Despite differences among endosomal TLRs, their expression has been detected both in different donors and in NK cell clones derived from the same individual [39]. As mentioned, TLR3 is overall highly expressed, followed by TLR7 and TLR8 moderately expressed and TLR9 expressed at low or undetectable level [27,30,31].

Additionally, studies demonstrate that mRNA expression of different TLRs is modulated upon different conditions. Examples are the overexpression of TLR3 in monocytes of Kaposi’s sarcoma patients [40] or the overexpression of TLR3, 4, 7, and 9 in tumor cells of patients affected by esophageal squamous cell carcinoma [41]. TLRs expression can be altered upon stimuli, such as granulocyte-macrophage colony-stimulating factor (GM-CSF), which induces the downregulation of TLR1, 2 and 4 in human monocytes, or IL12, able to increase the expression of TLR2 and TLR4 in mast cells [42]. Interestingly, it has been demonstrated that engagement of certain TLRs leads to the regulation of other TLRs expression. In particular, stimulation with TLR8 selective ligands induces both the upregulation of TLR2 and the downregulation of TLR7 and TLR9 in monocytes and in macrophages, respectively [32]. Our data demonstrate that mutual regulation of TLR expression exists also in NK cells. Similar to monocytes and macrophages, the main regulator of other TLRs expression is TLR8, whose triggering in NK cells induces the upregulation of TLR7 (unpublished data). Finally, recent evidence indicates that certain TLR functions differ based on the NK cell subset. In particular, our recent study demonstrated that TLR7 and TLR8 are both consistently expressed on freshly isolated CD56^bright^ and CD56^dim^ NK cell subsets, but only TLR8 agonists activate exclusively CD56^bright^ and not CD56^dim^ NK cell subsets [38]. Discovering how TLRs interact each other, regulating their own expression, could be useful in order to trigger TLR sequentially. The increased function and proliferation of NK cells induced in vitro by TLR ligands can be exploited to improve the infiltration of NK cells if they are administered intratumorally. The local therapy with TLR agonists is considered at present as a novel potential strategy to treat solid tumors. Furthermore, these compounds allow easy and quick amplification in vitro of NK cells which can be used as “off the shelf” cells to be adoptively administered in several tumors.

## 3. Endosomal TLR Agonists and Their Activity on NK Cells

### 3.1. TLR3 Agonists

TLR-mediated signaling pathways in NK cells are differentially regulated by TLR ligands and agonists. dsRNA produced by several viruses is known to stimulate TLR3 both on the cell surface and intracellularly, inducing NK cell cytotoxicity and production of CXCL10, IFN-γ and other inflammatory cytokines. In particular, upon stimulation, TLR3 interacts with the adaptor protein TRIF, also known as Toll-interleukin I receptor domain containing molecule 1 (TICAM), to induce the activation of interferon regulatory factor 3 (IRF-3) and nuclear factor kB (NF-kB) transcription factors that, in turn, activate the production of inflammatory cytokines. TLR3 can be also triggered by different synthetic molecules, including polyinosinic polycytidylic [Poly (I:C)] acid and its analogs. Poly (I:C) is a synthetic analogue of dsRNA acting as an agonist, not only of TLR3, but also of retinoic acid-inducible gene RLRs, that, once triggered, in turn regulates the adaptive immune system [43,44,45,46,47]. Interestingly, a cell-associated form of Poly (I:C) results in being more effective in activating TLR3 compared to soluble dsRNA, suggesting that dsRNA from dying cells is a more puissant and functionally pertinent TLR3 ligand than dsRNA from living cells [48].

Two analogues of Poly (I:C) have been designed with the aim to reduce the toxicity related to poly (I:C) administration: Poly-IC12U and Poly-ICLC are characterized by a high molecular stability and resistance to nucleic acid hydrolysis, respectively [49,50].

A second synthetic molecule triggering TLR3 is RGC100, a 100 bp long dsRNA which is characterized by a high solubility caused by its chemical structure [51] and serum stability, due to the 100% CG content [52]. These characteristics play a role in reducing the potentially toxic effects that are caused by other TLR3 agonists, such as Poly (I:C) [53].

A TLR3 agonist widely used in clinical practice is ARNAX, a DNA–dsRNA hybrid compound containing dsRNA (sequence of measles virus) so that it does not induce RNAi in human transcripts. This ligand is specific for TLR3 triggering the TICAM-1 pathway only. Notably, its conjunction sites of DNA–RNA and dsRNA regions show resistance to serum nucleases [54]. Compared to Poly (I:C), ARNAX induces poor inflammatory IFN-β and cytokine production in a TLR3–TICAM-1-dependent manner, indicating that the TLR3–TICAM-1 pathway causes a not significant and localized release of cytokines under priming DCs. Endosomal TLRs expressed by NK cells are also present in other innate immune cells, therefore the triggering of certain TLRs may lead to the simultaneous activation of different cell types. In particular, NK cells share a high expression of TLR3 with DCs; this contributes to promote the bidirectional crosstalk between each other occurring in the periphery or in secondary lymphoid tissues. Indeed, IL-12 released by TLR3-activated DCs renders NK cells responsive through TLR3 to dsRNA, increasing their anti-tumor/anti-viral cytotoxicity [55]. Moreover, NK cells promote the lysis of immature dendritic cells (DCs), thus favoring the selective survival of mature DCs [56]. Furthermore, TLR-stimulated NK cells increase pro-inflammatory cytokines secretion (TNF-α and IFN-γ), improving further DC maturation and subsequent induction of a type 1 immune response.

### 3.2. TLR7/8 Agonists

TLR7 and TLR8 are two endosomal receptors existing as a heterodimer (TLR7/8). TLR7 is characterized by having two binding sites: the first is devoted to interact with small ligands and is conserved in both TLR7 and TLR8, and the second site differs from that of TLR8 and is used to bind ssRNA and enhances activation of the first site [57]. While TLR8 is triggered by ssRNA AU- and GU-rich sequences [58], TLR7 is only activated by GU-rich sequences [59]. TLR7/8 agonists include resiquimod (R848), a small-molecular-weight synthetic compound belonging to the imidazoquinoline family and other molecules specific for either TLR7 or TLR8. Among those specific for TLR7 are imiquimod, also called Aldara or R-837, whose structure is similar to an adenosine nucleoside, gardiquimod, which is similar to imiquimod, with which it shares an imidazoquinoline structure, but it has stronger properties than the latter. An additional synthetic molecule related to imiquimod is 852A, described as a more potent and selective TLR7 agonist than imiquimod [60]. Different from imiquimod are: i. loxoribine, a guanosine analogue derivatized at position N7 and C8 (7- allyl-8-oxoguanosine) that enhances NK cells activity and induces production of cytokines such as IFNs [61], ii. bropirimine (2-amino-5-bromo-6-phenyl-4-pyrimidinone), an orally administered modulator that induces production of cytokines including IFN-α [62], iii. GS-9620, a potent and selective oral TLR7 that manifested a strong antiviral activity [63,64,65] and iv. SC1, a small molecule agonist of TLR7, stimulating in particular NK cells and mediating an efficient immune response. SC1 also showed an effective anti-metastatic activity in vivo [66]. Other TLR7 specific agonists include 3M-052, 3M-011, DSR-6434, and SZU-101, all showing anti-tumor activity in human cancer models [67]. Despite several molecules have been described to selectively trigger TLR7, the majority of which are used in clinical trials, only few molecules are currently available for TLR8 selective stimulation. VTX-2337, firstly described in 2012, is an agonist that activates NK cells to produce IFN-γ to increase the cytotoxicity of NK cells against target cell lines and antibody-dependent cell-mediated cytotoxicity (ADCC) by rituximab and trastuzumab [68]. Lastly, GS-9688 (selgantolimod) has been described as a potent orally active TLR8 agonist.

Similar to TLR3, TLR7/8 is expressed by both NK cells and conventional DCs. The latter, upon TLR7/8 stimulation, undergoes multiple functional changes, including maturation and expression of co-stimulatory markers, such as CD80 and CD86, and secretion of proinflammatory cytokines, including IL-12, TNF-α, and IL-6, which in turn act on NK cell activation [69].

### 3.3. TLR9 Agonists

Plasmacytoid dendritic cells (pDCs) share with NK cells also the expression of TLR9, recognizing unmethylated 2′-deoxyribo CpG DNA motifs in bacteria and viruses [70]. Its stimulation of pDCs induces IFN-α release, further supporting the activation of TLR9-responsive NK cells. 

Three distinct types of synthetic oligo-deoxy-nucleotides (ODNs) containing unmethylated CpG dinucleotides have been synthetized: i. A-type ODNs, ii. B-type ODNs and iii. C-type ODNs. All ODNs activate pDCs and induce the release of inflammatory cytokines, such as TNF-α and IFN-α [71]. TLR9 recognizes also the insoluble crystal hemozoin, originated as a byproduct of detoxification after digestion of host hemoglobin by plasmodium falciparum [72]. Interestingly, the Killer immunoglobulin-like receptor 3DL2 (KIR3DL2), a NK receptor well-known for recognizing class I human leukocyte antigen (HLA), is also able to bind ODNs and to induce KIR3DL2 down-modulation from the cell surface and translocation to the endosome to deliver ODNs toTLR9 [73]. Since killer Ig-like receptors (KIRs) expression is restricted to NK cells, this mechanism of antigen presentation is not shared with other immune cells, representing a unique feature of TLR biology on NK cells. On the whole, several natural or synthetic endosomal TLR agonists impacting NK cell function are at present suitable for in vitro and in vivo studies on cancer immunotherapy.

## 4. Endosomal TLR Agonists Stimulating NK Cells in Cancer Immunotherapy (CIT) 

In the last few years, TLR ligands have been used as adjuvants in cancer therapy either alone or in combination with other therapeutic approaches to enhance immune response against tumors. 

Despite endosomal TLRs develop a pro-tumorigenic function when they are expressed in tumor cells, their engagement by specific agonists was demonstrated to modify tumor growth by various mechanisms, such as inducing apoptosis and potentiating the effects of chemotherapy. Therefore, TLR agonists are promising candidates to be employed in CIT [74], as we reported in Table 1.

Here we will summarize the latest strategies to improve NK cell function by using endosomal TLR agonists.

### 4.1. TLR3 Agonists in CIT

Poly (I:C) has been largely tested in vitro and in vivo to assess NK and cytotoxic T-cell (CTL) activity [75], with a consequent reduction of tumor masses in tumor bearing mice. Unfortunately, the Poly (I:C) dose necessary to obtain an adequate response provokes several side effects, such as fever, erythema, or life-threatening endotoxin-like shock, probably due to a cytokine storm induced by the reagent [97]. Poly (I:C) is not only specific for TLR3, but it can also trigger cytoplasmic receptors such as RIG-I and MDA5 [98], thus improving the inflammatory cytokine production, as IL-6, TNF-α, and IFN-β [44]. Based on these premises, it became necessary to synthetize new agonists specific for TLR3 in order to minimize the cytokine production. Among them, cM362-140 is a sODN-dsRNA conjugate able to activate the TLR3/TICAM-1 pathway in the DC endosome. cM362-140 resulted in promoting NK activation and CTL proliferation in melanoma and lymphoma mouse models, respectively. Importantly, cM362-140 triggers exclusively TLR3, provoking an anti-tumor response without causing a cytokine storm [76]. Similarly, TAARD, a synthetic disaccharide derivative of diphyllin, is able to directly trigger TLR3 on primary NK cells and activate the STAT3 pathway. In particular, TAARD increases IFN-γ production by NK cells, and presents an additive effect in combination with IL-12 or IL-15 [77]. 

The use of anti-programmed death receptor 1 (PD-1) and anti-cytotoxic T lymphocyte–associated antigen 4 (CTLA4) monoclonal antibodies (mAbs) is often associated with conventional chemotherapy and radiotherapy treatments to improve patients’ survival. Nevertheless, a large proportion of patients still remain unresponsive to immune checkpoint treatment (ICT), but the association with TLR agonists improves ICT efficacy [99].

In this case, responsive mesothelioma and renal murine tumors expressed higher amounts of inflammatory genes, downregulated IL-10RA gene expression and increased activation of signal transducer and activator of transcription 1 (STAT1). An example is the administration of anti-PD1 and anti-CTLA4 mAbs in non-responsive cancer animal models pretreated with IFN-γ, whose efficacy are increased by the treatment with anti-IL-10 mAb and Poly (I:C). In this context, it has also been demonstrated that TLR3 triggering induces STAT1 phosphorylation, leading to higher IFN-γ production by circulating CD335^+^, CD11b^+^ and KLRG1^+^ NK cells recruited in TME [100]. Another TLR3-specific RNA agonist that improves the NK cell recruitment and function in TME when associated with anti-PD1 mAb is ARNAX. This compound upregulates cell adhesion genes, such as *Vcam1*, and downregulates inhibitory receptors genes, such as *Klra1* and *Tnfsf4* [54]. Reovirus-activated NK cells were efficiently employed in combination with cetuximab in order to improve ADCC for colorectal cancer treatment. This also suggests that reovirus activates TLR3 signaling pathway directly on NK cells [101]. An emergent concept is that different TLR agonists could be combined with each other with the aim of overcoming the suppressive TME in a more efficient manner. TLR9 and TLR3 agonists, administered locally by aerosol, were used in combination with mAbs against both Ly6G and Ly6C in order to deplete myeloid derived suppressor cells (MDSC) in lung metastases and to generate a more permissive TME for NK cell activation [102,103].

### 4.2. TLR 7/8 Agonists in CIT

R848 is the best known TLR7/8 agonist used for CIT. Cheadle et al. treated CD20^+^ lymphoma-bearing mice with obinutuzumab plus R848 and obtained the increase of NK cell function as activation, cytokine release, and ADCC [78]. Despite several positive results in terms of immune activation against cancer obtained with R848, its application is limited by the high toxicity and solubility, with a limited systemic employment [79]. Based on this consideration, it is urgently needed to synthesize new specific TLR 7/8 agonists characterized by low toxicity and increased systemic stability.

MEDI9197 is a dual TLR 7/8 agonist linked with a lipid tail in order to reduce solubility and improve retention at the injection site. MEDI9197 increases CD25 expression on NK cells and improves their cytotoxicity [81]. Moreover, the combination of cetuximab with small synthetic compounds able to trigger both TLR7 and TLR8 in a dose dependent manner induced tumor growth inhibition and increased NK cell-mediated ADCC in a lung murine model [104]. ADCC was also improved by using the T7-MG7 compound that is made up by a small TLR7 agonist (T7) with a gastric cancer antigen (MG7) and is characterized by low toxicity and long-term effect [87].

Since one of the main strategies adopted by tumors to escape from immune-surveillance is to downregulate MHC class I expression, Doorduijn et al. used imiquimod (TLR7 ligand) to promote an interplay between CD4^+^ T cells and NK cells, with a consequent regression of MHC-I^low^ tumors. In this case, imiquimod, commercially known as Aldara (a FDA- and EMA-approved cream with a limited systemic toxicity), was used in combination with IL2. They observed an increased infiltration of activated CD69^+^ NK cells into the tumor site. In addition, they found that NK cells, along with CD4^+^ T cells, are able to control the residual tumor lesions after Aldara treatment [83]. Imiquimod was also proposed to prolong tumor growth control by NK cells in B-Raf proto-oncogene (BRAF) mutated melanoma models treated with BRAF inhibitors [84]. SC1, another TLR7 agonist, induces a potent TLR7-mediated anti-tumor immune response with a lower toxicity compared to R848 [66]. This molecule was successfully used to revert anergy of infiltrated NK cell in RMA-S lymphoma bearing mice; in this model, the down regulation of Ly49A and CD96 inhibitory receptors and up-regulation of CD69 activation marker, IFN-γ release, and cytotoxicity by NK cells were observed [86].

Recently, many approaches were proposed to selectively stimulate TLR8. These include the use of VTX-2337, which specifically triggers TLR8 more efficiently than R848 or 3M-002 (CL075) [88]. Importantly, it was demonstrated that TLR8 agonist promotes the NK cell-dependent immune response against tumors more efficiently than other endosomal TLRs [90]. Several in vitro studies, demonstrated that NK cell function modulated by TLR8 agonists improves the efficacy of already approved therapies with monoclonal or polyclonal antibodies [105]. In many cases, after VTX-2337 treatment, an increased production of IFN-γ mediated by NK cells was observed. In particular, TLR8 triggering promotes cleavage of caspase-1, which mediates the release of IL-1β and IL-18, improving IFN-γ production by NK cells [68,88]. Recently, we demonstrated that VTX-2337, but not other specific TLR7 agonists, such as imiquimod, loxoribin, or modified 8-hydroxy-adenine, activated exclusively CD56^bright^ NK cell subsets with high release of IFN-γ and TNF-α and increase of cytotoxic activity [38]. Despite the extensive literature describing the ability of TLR7/8 agonists to promote NK cell function, none of them showed the activity of TLR8 ligands on these cells. Furthermore, we observed the TLR8-mediated activation of NK cells derived from the ascitic fluid of patients with metastatic ovarian carcinoma, mainly represented by CD56^bright^ NK cell subsets. In line with our data, Amin O. et al. showed that GS-9688, another TLR8 specific agonist used in the treatment of patients with chronic hepatitis-B, increases NK cell frequency and TRAIL expression on CD56^bright^ NK cells subset [89].

### 4.3. TLR9 Agonists in CIT

In the last decade, it was reported that TLR9 ligands can directly trigger TLR9 on NK cells [73]. TLR9 agonists are mostly used in combination with inhibitory checkpoint antibodies, as anti-PD1 mAb, in non-responding cancer patients [106]. A study analyzed the activity of HP06T07I, a new ODN TLR9 agonist, to promote NCR1^+^ NK cell infiltration in a colon carcinoma mouse model [90]. NK cell recruitment in non-small-cell lung cancer (NSCLC) model was also obtained by using litenimod (Li28), a TLR9 agonist, in combination with the immunotherapeutic vaccine based on modified vaccinia virus Ankara (MVA), namely TG4010 [91]. In addition, it was demonstrated that JAK1/JAK2 mutation in melanoma patients, which provokes resistance to anti-PD1 mAb treatment, can be overcome with the treatment of SD-101 TLR9 agonist. After treatment, it was possible to detect the NK cell recruitment in the tumor site, with improved ability in controlling tumor growth [92]. MGN1703 effect was evaluated in many infectious diseases, such as HIV-1 infection. This compound acts especially on CD56^dim^ NK cells, in which it improves degranulation activity, IFN-γ production, NKp46 and NKG2A receptor expression [93].

## 5. Clinical Trials with Endosomal TLR Agonists Stimulating NK Cells 

At present, 62 clinical trials (active, recruiting, completed, or terminated) on the TLR agonist employment for CIT are registered, and most of them involve the endosomal TLR agonists in combination with other treatments (chemotherapy, radiotherapy, or mAbs treatment)(https://clinicaltrials.gov). The VTX-2337 molecule (USAN: motolimod) has been used in 11 clinical trials against solid tumors, such as ovarian and squamous cell carcinoma of the head and neck (SCCHN). In particular, the studies of Chow L. and Dietsch G. N were focused on VTX-2337 in combination with cetuximab for the treatment of SCCHN patients in two phase II clinical trials (NCT01836029, NCT01334177); in this context, they observed an increased NK cell-mediated ADCC [88,107]. Other studies employed MGN1703 for treatment of colorectal carcinoma in combination with chemotherapy treatment. They evaluated especially the frequencies of NK and NKT cells as positive biomarkers in colorectal carcinoma patients (NCT01208194) [94]. Therapeutic advances in childhood leukemia (TACL) and lymphoma phase I consortium performed a pilot study on three patients with minimal residual disease (MRD) positive acute leukemia (NCT01743807) by using GNKG168 as a TLR9 agonist. They demonstrated that the *SIGIRR, IL1RL1, CCR8, IL7R, CD8B*, and *CD3D* genes are downregulated after GNKG168 treatment. In particular, IL7R decrease could promote CD56^bright^ NK cell subset suppression of graft-versus-host disease (GvHD) in acute leukemia patients [95]. Nevertheless, most of clinical trials didn’t achieve the desired therapeutic effect and provoked different adverse reactions. For example, MEDI9197 used as an adjuvant for advanced solid tumor treatment (NCT02556463), induced side effects such as cytokine release syndrome [104]. The description of clinical trials using endosomal TLR agonists in human tumors has been summarized in some complete, recent reviews [11,82,108,109,110].

## 6. Novel Approaches of CIT with TLR-Activated NK Cells

Targeted drug delivery can significantly influence the efficacy of the molecule in terms of pharmacokinetics and bioavailability [111]. In the past decades, diverse drug-delivery technologies, including nano- and microparticles, co-crystals, and microneedles have been developed to maximize therapeutic efficacy and minimize unwanted side effects of therapeutics. In particular, vaccine development is often accompanied by adjuvants that promote strong T cell responses by prolonging antigen presentation by DCs and by activating NK cells [112]. Aluminum salts are the most widely applied adjuvants in human vaccines since they are safe and well tolerated [113]. The micrometer-sized aluminum aggregates were transformed into nano-sized vaccine carriers by shielding their positive charges with a polyethylene glycol (PEG)-containing polymer [114]. Internalization of these molecules is highly dependent on scavenger receptor A-mediated endocytosis [114]. Polymeric nanoparticles (NPs) and liposomes have been investigated to encapsulate payloads, including drugs, proteins, vectors, and nucleic acids [115]. Moreover, NPs tend to accumulate in tumors and not in normal tissue as a consequence of leaky tumor vasculature and damaged lymphatic drainage [116]. 

In the last few years, nanomaterials have been designed to boost NK cell CIT. In particular, most NPs were developed to modify the immunosuppressive TME [117] so as to improve the recruitment of NK cells in tumor sites [101,118] and to restore NK cells’ function by increasing their proliferation, cytotoxicity and cytokine production [119]. Many studies assessed NP loaded with TLR agonists, alone or in combination with other molecules [120]. For example, PLGA-NP co-loaded with indocyanine green (ICG) fluorescent dye and R848 (PLGA-ICG-R848) was proposed for treatment of prostate cancer, where it induced an increase of cytotoxic activity of NK cells [80]. Kim, H. et al. used a pH-responsive polymeric NP wherein they encapsulated a TLR7/8 agonist [121]. The treatment with this compound prolonged NK cell activation and improved in vivo cytotoxicity much more than using a soluble agonist. In addition, TLR7/8 agonist-loaded NPs potentiated NK cell mediated ADCC in combination with cetuximab in melanoma model [122]. Other groups used a virus-like particle (VLP) CMP-001 as a ligand for TLR9. The VLP is composed by CpG-A ODN incapsulated in a Qβ bacteriophage capsid that protects the ligand from a rapid degradation. CMP-001 was employed in gastrointestinal and pancreaticobiliary cancer, thus improving NK cell infiltration and activity in vivo [123]. Importantly, CMP-001 can directly trigger TLR9 on NK cells, improving the anti-PD1 mAb treatment in melanoma mouse models [96]. Perry, J. L. et al. utilized nanoparticle replication in nonwetting templates (PRINT) to deliver TLR9 agonists into murine lungs. They observed not only an indirect stimulation of NK cells mediated by DC and macrophages, but also a direct interaction between PRINT-CpG and NK cells, suggesting a direct TLR9 triggering of these cells [124].

The employment of NP-based strategies for NK cell immunotherapy starts to be explored, and the majority of recent data are in favor of the combination of endosomal TLR agonists and NPs [82,125,126,127,128]. 

## 7. Conclusions and Future Perspectives

TLRs are expressed on tumor cells and immune cells of the TME. Endosomal TLRs, which recognize exogeneous (pathogens) and endogenous DNA and RNA, are mainly expressed on APC, while their expression on NK cells has been long debated in the past. Recently, we definitively demonstrated that the four endosomal TLRs (TLR3, 7, 8, and 9) were consistently expressed on freshly isolated CD56^bright^ and CD56^dim^ NK cell subsets through different genetic and biochemical techniques. By checking TLR7 and TLR8 for protein expression and endosomal localization, we observed that TLR8, but not TLR7, mostly localizes at the late endosome level. In addition, taking advantage of synthetic specific agonists of human TLRs, we demonstrated that CD56^bright^, and not the CD56^dim^ NK cell subset, is responsive to TLR8 engagement, while TLR7 is not [38]. Thus, these TLR ligands may be considered as new relevant immunotherapeutic adjuvants to increase treatment efficacy and improve cancer patient outcomes. Indeed, several in vitro and in vivo results suggest that TLR8 agonists deeply modify TME, since they may: i. reverse the suppressive functions of human tumor-derived CD4^+^ or CD8^+^αβ T cells, and γδ T cells [85,129,130]; ii. induce the switch from M2 to M1 profile of intra-tumor macrophages [131]; iii. enhance apoptosis of MDSC [132]; iv. prevent T cell senescence/exhaustion [133]; and v. induce in vivo the metabolic control of CD4^+^ T regulatory cells in the solid tumor TME (ovarian cancer) [134]. Several data on murine models and clinical trials strongly suggest that locally infused TLR7/8 agonists delayed the tumor growth and induced tumor regression [81,107,135,136,137]. Thus, TLR8 agonists can be considered a novel strategy able to promote anti-tumor immunity activating NK cells, which, in turn, prime robust and sustained adaptive immune response against the tumor. 

However, many more studies are necessary to better understand the role of TLRs locally administered in cancer TME, taking into account several variables, such as TLR expression on tumor cells, mutagenesis, roles of TLR adaptors, and many other related mechanisms. In addition, it is important to underline that each cell population has different TLR expression and responds differently to TLR agonists. 

Whereas TLR ligand may be used in DC vaccines which are widely assayed for CIT, immunotherapy with TLR-activated NK cells is still under investigation. In recent years, nanomedicine has offered new strategies for CIT. One of the advantage of NPs is that they can be designed to various sizes, shapes, and functions. They can be modified to target specific components in TME, molecules on innate immune cell surfaces, or loaded with various drugs or adjuvants, thus achieving targeted delivery and simultaneous delivery of therapeutic agents. Nano- and micro-particle systems bound to TLR agonists allow to use a lesser dose of adjuvants and increase anti-tumor immunity compared to traditional treatment. For instance, the development of biodegradable PLGA-PEG NPs, a delivery vehicle for local, slow and sustained release of Poly (I:C) and R848, are potent candidates to treat solid tumors resistant to first-line therapies [80]. Despite this, modified NPs are already used to target DC. Many questions about their toxicity and immunogenicity still remain open. 

NPs loaded with TLR agonists could be proposed to selectively trigger NK cells in adoptive settings. As previously reported, the triggering of TLR8 is a good strategy to active CD56^bright^ NK cells [38]. Thus, a novel therapeutic strategy could use NPs loaded with TLR8 agonists as an innovative procedure to activate and expand in vitro circulating or intra-tumoral CD56^bright^ NK cells, which will be later used for adoptive therapy in autologous or allogenic recipients (Figure 1).

## Figures and Tables

**Figure 1 biomedicines-11-00064-f001:**
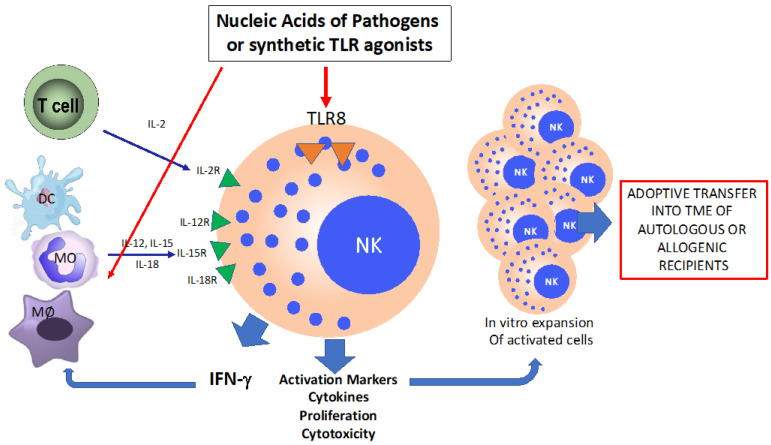
Schematic representation of direct NK cells activation through TLR8 engagement and indirect activation by accessory cells through cytokines.

**Table 1 biomedicines-11-00064-t001:** Immunologic effects of endosomal TLR agonists used as cancer vaccine adjuvants in CIT.

TLRs	TLR Agonists	Immunologic Effects	Potential Clinical Use	Ref.
*TLR3*	Poly-I:C	inflammatory cytokine production; tumor mass reduction	mesothelioma and renal tumors	[75]
	Poly-IC12U	upregulation of costimulatory molecules in DCs	-	[49]
	Poly-ICLC	shift to Th1 profile with the induction of inflammatory cytokine and chemokines	ovarian cancer	[50]
	RGC100	activation of DCs by targeting endosomal TLR3 with low toxic effect.	-	[51,52]
	ARNAX	recruitment of NK cell into TME in association with anti-PD1 mAb	tumor expressing high levels of PD1	[54]
	cM362-140	induction of NK activation and CTL proliferation	melanoma and lymphoma	[76]
	TAARD	increase in NK cells of IFN-γ production in association with IL-12 or IL-15	-	[77]
*TLR7/8*	Resiquimod	increase in NK cell of cytokine release and ADCC in association with obinutuzumab	prostate cancer	[78,79,80]
	MEDI9197	increase of CD25 expression and toxicity on NK cells. In association with cetuximab it decreases tumor growth	lung cancer	[81,82]
*TLR7*	Imiquimod	interplay between CD4^+^ T cell and NK cells; increase in CD69^+^ NK cells infiltration; extension of tumor growth control	BRAF mutated melanoma	[83,84]
	Gardiquimod	-	-	-
	852A	induction of PBMC to secrete IFN-α and CXCL10	refractory solid tumor	[60,85]
	Loxoribine	enhancement of NK cells activity and of IFNs production	advanced solid tumor	[61]
	Bropirimine	IFN-α production	bladder tumor	[62]
	GS-9620	antiviral activity	-	[63,65]
	SC1	inversion of anergy of infiltrated NK cell	lymphoma	[86]
	3M-052	inhibition of both local and systemic tumor growth	melanoma	[67]
	3M-011	increase in the antigen-presenting activities of DC	pancreatic cancer and colon carcinoma	
	DSR-6434	reinforcement of the effect of radiation therapy	colon cancer and renal cell carcinoma	
	SZU-101	-	breast cancer, T cell lymphoma and gastric cancer	
	T7-MG7	improvement of ADCC	gastric cancer	[87]
*TLR8*	Motolimod	Increase of IFN-γ production from NK cells; ADCC increase with cetuximab	ovarian cancer and SCCHN	[68,88]
	Selgantolimod	increase of NK cell frequency and TRAIL expression on CD56^bright^ NK cells subset	-	[89]
	3M-002	DC release of IL-12(p70)	-	[88]
*TLR9*	ODN-A, ODN-B, ODN-C	pDCs activation with TNF-α and IFN-α release	glioblastoma	[71]
	HP06T07I	increase of NCR1^+^ NK cell infiltration	colon carcinoma	[90]
	Litenimod (Li28)	enhancement of NK cell recruitment	NSCLC	[91]
	SD-101	promotion of NK cell recruitment	melanoma	[92]
	MGN1703	improvement of degranulation activity	Colorectal carcinoma	[93,94]
	GNKG168	improvement of NK cell infiltration and activity	acute leukemia	[95]
	CMP-001	direct activation of NK cells	gastrointestinal and pancreatico-biliary cancer	[96]

## Data Availability

Not applicable.

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
