# Peer review of "The Latest Approach of Immunotherapy with Endosomal TLR Agonists Improving NK Cell Function: An Overview"

_biomedicines, 2022, doi:10.3390/biomedicines11010064_

Round 1

Reviewer 1 Report

The manuscript summarized recent progress on TLR ligands mediated cancer immunotherapy (CIT), which mainly focus on the application of nature killer (NK) cells in tumor immunotherapy. The review paper were well organized and documented, yet few things need to be done to improve the manuscript.

1. The manuscript need to be well edited and the text should be polished, some sentences are really hard to understand.

2. Bunches of abbreviations were used in the paper, but some weren't explained, like NCR and KIR.

3. The augmented function (activation) and increased infiltration of NK cells by TLR ligands should be distinguished and discussed in the paper, because it seems the lymphocyte recruitment (increased number) playing an important role in CIT.

4. For drug delivery, Lipid nanoparticles are recently well used system, whether it is also used in TLR ligands mediated CIT, otherwise the potential application can be talked.

Author Response

The manuscript summarized recent progress on TLR ligands mediated cancer immunotherapy (CIT), which mainly focus on the application of nature killer (NK) cells in tumor immunotherapy. The review paper were well organized and documented, yet few things need to be done to improve the manuscript.

Comment 1.1. The manuscript needs to be well edited and the text should be polished, some sentences are really hard to understand.

Reply: The manuscript has been accurately revised and re-edited by a native English speaker, the text has been polished by refuses and some sentences have been rewritten for a better understanding

Comment 1.2. Bunches of abbreviations were used in the paper, but some weren't explained, like NCR and KIR.

Reply: All abbreviations used in the paper have been explained in the text of the revised version

Comment 1.3. The augmented function (activation) and increased infiltration of NK cells by TLR ligands should be distinguished and discussed in the paper, because it seems the lymphocyte recruitment (increased number) playing an important role in CIT.

Reply: We would like to thank the reviewer for the suggestion. In the last version of the review, we have already highlighted the efficacy of TLR agonists to improve NK cell infiltration in tumor environment. However, we additionally stressed in the new version the concept of increased NK infiltration and the strategy to quickly expand NK cells in vitro for new adoptive therapy of cancer. (see pag 4, lines 143-149).

Comment 1.4. For drug delivery, Lipid nanoparticles are recently well used system, whether it is also used in TLR ligands mediated CIT, otherwise the potential application can be talked.

Reply: As underlined by the reviewer, lipid nanoparticles loaded with TLRs agonists may highly contribute to improve CIT. For this reason, we dedicated a section in the paper (paragraph 6 entitled: “Novel approaches of CIT with TLR-activated NK cells”), in which we summarize the latest nanoparticles approaches employed to improve NK cell function in CIT. In particular, nanoparticles loaded with TLR7, TLR8 or TLR9 agonists are used to efficiently enhance NK cell cytotoxicity or ADCC. In the new version several new references on this topic have been quoted [1-4] [5] (see pag 17, line 679 and pag 18, line 718-725)

Thank you very much for accepting the revised version.

With my best regards

Prof. Enrico Maggi, corresponding author

Reviewer 2 Report

Most of this review is a description of the structure and functions of TLR in several cell types. The interesting portion of the paper is based on recent findings of the AA about TLR7 and 8 expression on CD56bright and  CD56dim NK cells. Given their preliminary findings, the AA anticipate a possible role of endosomal TLR in increasing NK function in immunotherapy.

However, the expression of endosomal TLR is still under debate and this represents a limitation of the review. I would encourage the AA to provide more experimental work about presence, regulation and function of endosomal TLR in NK cells.

Author Response

Most of this review is a description of the structure and functions of TLR in several cell types. The interesting portion of the paper is based on recent findings of the AA about TLR7 and 8 expression on CD56bright and CD56dim NK cells. Given their preliminary findings, the AA anticipate a possible role of endosomal TLR in increasing NK function in immunotherapy.

Comment 2.1 However, the expression of endosomal TLR is still under debate and this represents a limitation of the review. I would encourage the AA to provide more experimental work about presence, regulation and function of endosomal TLR in NK cells.

Reply: The expression of endosomal TLR on NK cells has been long under debate, mainly for the lack of availability of valid specific antibodies. However, at present, data on the presence of TLRs (and of endosomal TLRs in particular) on NK cells are more solid than in the past.  In human NK cells, TLR1 mRNA levels are the highest, followed by TLR2, TLR3, TLR5, and TLR6 mRNAs at moderate levels, while TLR 7/8 and 9 mRNA expression levels were low [6-9]. Along the last two decades the group of Alessandro and Lorenzo Moretta highlighted and repeatedly confirmed the  presence of all endosomal TLRs on NK cells by different methods (mRNA expression, Western Blot analysis and confocal microscopy) performed on purified NK cells, NK92 cell line or NK cell clones [10-14]. Finally, the presence of endosomal TLRs has been directly demonstrated at the protein level and functionally through the activation of purified NK cells by specific ligands/agonists, particularly of TLR8 [15]. These concepts have been added to the text of the new version (see pag 3, lines 113-119).

Thank you very much for accepting the revised version.

With my best regards

Prof. Enrico Maggi, corresponding author

Reviewer 3 Report

I carefully read and appreciated this review.  The authors accompany the reader in understanding the use of TLR antagonists to develop new therapeutic approaches in oncology by focusing our attention on their personal experience and competence.

 I suggest the authors improve the introduction by briefly describing the role of the various types of TLRs in different pathologies starting from the first clinical evidence to improve the comprehension of the critical role of TLRs in the immunosurveillance deficit during the neoplastic progression.

Sometimes, focusing too much on the future makes us lose the memory of the past. Diving into relevant details from past investigations more than once helped us better plan future scientific directions.

Author Response

I carefully read and appreciated this review.  The authors accompany the reader in understanding the use of TLR antagonists to develop new therapeutic approaches in oncology by focusing our attention on their personal experience and competence.

Comment 3.1. I suggest the authors improve the introduction by briefly describing the role of the various types of TLRs in different pathologies starting from the first clinical evidence to improve the comprehension of the critical role of TLRs in the immunosurveillance deficit during the neoplastic progression. Sometimes, focusing too much on the future makes us lose the memory of the past. Diving into relevant details from past investigations more than once helped us better plan future scientific directions.

Reply: Since TLR are expressed in many types of cells, they can potentially play a role in different type of pathologies, from allergies to autoimmunity and cancer. However, there are not evidences of clear cut association of single TLR with different diseases.  Indeed, TLR signaling can be often influenced by other cell signals (as soluble factors or cell-to-cell contact) that modify the TLR effects in different immune-mediated diseases. In this review, we focused on the TLR expression and function in NK cell subsets. In particular, since the involvement of NK cells in anti-tumor response is widely documented, we reported the latest experimental studies and clinical trial in which endosomal TLR agonists are used to improve NK cell function against tumor progression.

However, for completeness, in the new version we implemented the introductory section by shortly describing the known association of TLRs and their agonists in the main infectious diseases (see pag 2, lines 64-77). In addition, the dual role of TLRs in the immunosurveillance against tumors, which may impair or improve neoplastic progression, has been also reported (see pag 2 and 3, lines 80-89)

Thank you very much for accepting the revised version.

With my best regards

Prof. Enrico Maggi, corresponding author

Round 2

Reviewer 2 Report

I confirm my previous comment about this paper: "the expression of endosomal TLR is still under debate and this represents a limitation of the review. I would encourage the AA to provide more experimental work about presence, regulation and function of endosomal TLR in NK cells".

Author Response

We thank the reviewers for their comments and inputs aimed at improving our manuscript.

The expression of endosomal TLR in NK cells is no longer under debate since many supporting evidences have been provided over the past years. These include:

•Chalifour, A., et al., Direct bacterial protein PAMP recognition by human NK cells involves TLRs and triggers alpha-defensin 542 production. Blood, 2004. 104(6): p. 1778-83.  
•Sivori, S., et al., CpG and double-stranded RNA trigger human NK cells by Toll-like receptors: induction of cytokine release and 553 cytotoxicity against tumors and dendritic cells. Proc Natl Acad Sci U S A, 2004. 101(27): p. 10116-21. 554  
•Lauzon, N.M., et al., The direct effects of Toll-like receptor ligands on human NK cell cytokine production and cytotoxicity. Cell 544 Immunol, 2006. 241(2): p. 102-12.  

Where authors showed that purified human NK cells expressed mRNA for TLR1–10 and were directly responsive to most TLR ligands. In particular, treatment of NK cells with TLR ligands resulted in an increase in their IFN-γ production and/or cytotoxic activity.

In addition, in a recent paper we clearly demonstrated both the mRNA expression of endosomal TLRs in NK cells and the presence and localization of the relative proteins by Western Blot analysis and confocal microscopy (Veneziani, I., et al., Toll-like receptor 8 agonists improve NK-cell function primarily targeting CD56bright CD16- subset. J 555 Immunother Cancer, 2022. 10(1). )

Clinical trials based on endosomal TLRs agonists stimulating NK cells further corroborate that the expression of endosomal TLRs in NK cells is currently widely accepted.

With best regards,
Dr. Claudia Alicata and Prof. Enrico Maggi